# Evaluation of a Road Safety Awareness Campaign Deployed along the Roadside in Saguenay (Québec, Canada)

**DOI:** 10.3390/ijerph20116012

**Published:** 2023-05-31

**Authors:** France Desjardins, Martin Lavallière

**Affiliations:** Department of Health Sciences, Université du Québec à Chicoutimi, Chicoutimi, QC G7H 2B1, Canada; f1desjar@uqac.ca

**Keywords:** primary prevention, evaluation, responsible behaviors, road safety, built environment

## Abstract

For the past few years, police officers from the City of Saguenay have been installing a billboard combined with a damaged car along roadsides to make drivers aware of the road risks related to dangerous behaviors at the wheel. To assess the short-term effect of this device, evaluative research with a quasi-experimental design with pre-exposure, during, and post-exposure. The results show a significant decrease (*p* < 0.001) of 0.637 km/h for the first site (a 70 km/h zone) and 0.269 km/h for the second site (a 50 km/h zone) when the device is exposed. At the time of this last evaluation, a reduction of 1.255 km/h remained even after the advertising panel was removed. Although minimal, this speed reduction where the billboards are placed shows the police that this awareness-raising approach works since it reduces the speed of motorists at very low cost.

## 1. Introduction

In Quebec, in 2021, more than 27,000 people will have suffered minor to fatal injuries [1]. Speed has been identified as one of the contributing factors in these collisions [2,3]. This worrying situation is mobilizing all police forces, especially in the City of Saguenay. Over the span of four years, the number of speed-related offenses has increased steadily from 14,649 (2016) to 18,151 (2019) [4], for a population of around 140,000 people living in the City of Saguenay’s territory.

In 2017, the Saguenay Police Department created and deployed an awareness campaign on the consequences of speeding while driving, according to the financial means at its disposal, to combat this increase in speed-related offenses on its territory. The police organization considered this situation worrying since it has been recognized that a reduction in the average speed of only 1.6 km/h can generate a 5% reduction in collisions [5].

However, respecting speed limits is connected to the general safety of the population, including other drivers and vulnerable users such as pedestrians and cyclists. The scientific literature offers solutions for reducing speeding such as the improvement of infrastructure [6,7,8,9], the use of automated speed camera enforcement [10,11], and an increased police presence [12,13,14,15]. Unfortunately, the City of Saguenay police department did not have access to these solutions due to budgetary constraints and limited deadlines. In these circumstances, the present research question was: How effective is the awareness campaign deployed by the Saguenay Police Department in reducing speeding among the general population? but especially with those who do not respect the speed limits?

To answer this question, this article presents a review of the literature to address the targeted group and the communication strategy of this awareness campaign. The associated research methodology is a quantitative single- and multiple-time series. Finally, the results will be presented and discussed to connect the main theoretical and practical contributions.

Our article exposes a double contribution to the driver education communication strategy in terms of the consequences related to speeding. The first contribution involves the immediate and short-term reactions of drivers who see the advertisement campaign using a damaged car on the side of the road. The second contribution refers to the effects of this awareness campaign on driving behaviors that do not respect the posted speed limits. Overall, the answer to the aforementioned research question will allow the Saguenay Police Department, as well as other organizations, to make an informed decision regarding the use of such an awareness tool.

## 2. Awareness Campaign

### 2.1. Target Group

According to several authors, the profile of drivers who take the most risks, such as speeding while driving, is that they are experienced male drivers. In fact, they emphasize that confidence in their past experiences influences their risk-taking [16,17]. For Wilde [18], drivers’ past experiences with speeding refer to a risk homeostasis. Under this reasoning, according to Noar [19] and Strecher et al. [20], a communication campaign must have a target audience since it is a success factor.

### 2.2. Communication Strategy

In 2020, the government of Quebec spent eight million dollars to promote road safety awareness among the general population [21]. Despite this fact, only excessive speed increases the probability of being the victim of a collision for most drivers [22]. This means that they do not think or believe that low and medium speeding excesses could also cause a collision. However, having a collision or getting a ticket is not necessarily required, as research has observed behavioral changes in the absence of these consequences [12,13,14,23,24]. A meta-analysis underlines that the effects of a mass advertising campaign on road safety are greater when the situation is linked to a legal aspect, as in the present case [25]. Furthermore, another meta-analysis highlights the relevance of using radio, television, or video instead of the written press to raise public awareness [26]. This campaign uses visuals that can be similar to television or video but are aligned along the roadside. This real-world communication strategy has been suggested by Freeman et al. [27]. In a real-world context, communication strategy refers to credible staging in a place where the situation could actually take place.

Unlike the Société de l’assurance automobile du Québec’s (SAAQ) communication strategy, which proposes advertising campaigns on TV media, in newspapers, and/or on billboards, the Saguenay Police Department has used an approach that involves a realistic setting and clearly outlines the potential consequences related to speeding (i.e., a crashed vehicle).

## 3. Methodology

To determine if this is an effective awareness campaign, the investigators and the City of Saguenay Police Department asked the following question: How effective is it in reducing the speed of automobile drivers? This question relative to evaluative research sets out a scientifically based approach to analyzing the adequacy between the various components of the awareness campaign and the very short-term effects of the latter observed at the vehicle display locations. Specific questions congruent with effectiveness [28] are: Do the effects observed in the short term correspond to the initial objectives set? How do the effects noted in drivers exposed to the crashed car compare to the speed before the car was exposed and following its removal? The expected results are a decision involved with the stopping, improvement, or continuity of this advertising campaign for the City of Saguenay Police Department, given their desire to implement actions based on evidence and to diligently allocate public funds [29]. Moreover, if the results show a desired effect at a low cost, the awareness campaign could be used by other police forces. Thus, the short-term effects of this mass advertising campaign will be scientifically documented.

Considering the innovative and unique aspect of the awareness campaign created by the police department, our research team proposed that they conduct evaluative research. Evaluative research should begin by producing a logical framework [28]. The logical framework sets the awareness campaign’s context, primarily the need, the relevance, and the objectives. Inputs are the resources mobilized to describe the processes or activities carried out. It concludes with the short-term effects. This framework makes it possible for the operational logic of the awareness campaign to be laid out. The following Figure 1 shows the framework’s different components (see next Figure 1).

For police officers, it is essential to raise public awareness of the potential consequences of speeding on the road network since collateral victims and vulnerable road users are the first victims of their inappropriate behaviors. Moreover, police officers are mandated to organize prevention activities beyond their coercive activities. This activity designed by the police required the purchase of wrecked cars ($500 per car), billboards ($300 each), and a tow truck ($1200). In addition to the purchase of this material, minor expenses are incurred for storing and transporting the vehicle to implement roadside advertising in the territory (see Figure 2).

This campaign tool has been developed to be affordable and easily movable across the territory covered by the organization (1136 km^2^). Built as a single piece of equipment, it facilitates its transportation, and the panel is used to hide the sun’s rays, which can come from behind and shine directly into the eyes of the person looking at the installation. The target population is those who come into contact with the campaign. The police also thought of using this tool during large gatherings when people are walking and generally using their cars at the end of the evening, since the territory is vast with public transportation and is mostly non-existent at this time of day. In these circumstances, the panel’s visibility increases, and it is also legible. Usually, the police place the wrecked car on the side of a very busy road for a few days and change its location to raise awareness among more people and create a surprise effect. However, the roads targeted by the police were busy and known to be problematic for speeding offenses. The general population, especially drivers who do not respect speed limits, is the group targeted by the exposure of the damaged car, mostly drivers aged between 18 and 60 years old.

In order to evaluate the effect of this campaign on driver behavior, the quantitative methodology is determined by a covariance analysis of the short-term effects in relation to the objectives mentioned above [28]. This evaluation’s primary hypothesis is:

**H1**: 
*Motorists do not reduce their speed when they see the crashed car.*


**H2**: 
*Motorists reduce their speed when they see the crashed car.*


The alternative hypotheses are:

**H3**: 
*Motorists who drive over the speed limit reduce their speed.*


**H4**: 
*Motorists reduce speed after car crash exposure.*


**H5**: 
*Drivers reduce their speed more when they are in a 50 km/h zone than when they are in a 70 km/h zone when the accident car is exposed (they have more time to see it).*


A semi-experimental before, during, and after design (without a control group) is proposed to answer this hypothesis. The dependent variable is the speed of the vehicles in kilometers per hour, and the independent variable is the presence or absence of the exposed crashed vehicle.

Our team wondered about the natural elements that can disrupt automobile driving and the fact that we saw the advertising campaign along the side of the road. The presence or absence of rain, the distance relative to visibility, the wind speed, and the illumination, since the trailer is not lit due to constraints in energy consumption and its positioning along the roadside, are control variables.

Monitoring includes different weather variables measured hourly by a nearby Environment Canada station; data can be found on the website for all hours. We also considered the direction of the cars, considering that the trailer was more visible on one side of the road, and the type of vehicle recorded by the radar (small car vs. truck). Just like Simpson, Frewing, and Bayer [30], these variables have negligible effects on vehicle speed; however, we have kept them in our analyses considering the potential seasonal effect a region like ours could present annually. The simple chronology of events is the measurement of the speed of the cars before and during exposure.

The chronology of events is the measurement of the speed of passing vehicles before, during, and after exposure to the campaign. The radar used is relatively small in terms of size and can be installed along the roads. It also provides its own electricity for autonomy via an on-board battery system (see Figure 3 below).

The radar was installed by road workers from the City of Saguenay. The speed data was recorded by an automated radar, and the employee sent us the results in an Excel file when it was picked up after project completion. In this file, the date, time, speed of each vehicle, direction, and type of vehicle detected by the radar (small car or truck) were noted.

### 3.1. Data Collection

Two tests in two different places are carried out for data collection.

The first test was carried out from June 23 to June 20, 2019. The pre-exposure phase took place from May 23 to May 30, 2019, where the statistical radar recorded the speed of all the vehicles on the segment evaluated. The first experiment was carried out in an area where the speed limit is 70 km/h (see Figure 4 for more details). We can see that the daily medians are relatively stable and are around 80 km/h, with recordings above 100 km/h.

Thereafter, the crashed car was installed from May 30 to June 20, and the radar continued to record data, thus allowing for a comparison of pre-exposure vs. exposure. Unfortunately, post-exposure data collection was not possible for this event due to administrative constraints from the municipality. For this reason, and considering H4 and H45, the research team decided to do a second test. The second site chosen was about twenty kilometers away to try to get new people.

The second test was carried out from September 17 to November 6, 2019. The pre-exposure phase took place from September 17 to October 9, the exposure phase from October 10 to October 30, and the post-exposure phase from October 31 to November 6, 2019. For the second test, it was carried out in an area where the speed limit is 50 km/h (see Figure 5 for more details).

We can observe that the speeds recorded during both pre-tests are mainly around the prescribed limit, and the medians are stable.

### 3.2. Data Analysis

We inputted all results from the Excel files into the SPSS software suite (IBM, Armonk, NY, USA, Version 27). Each radar data entry was enhanced with control data: visibility, rain, wind, sunlight, vehicle type, and traffic direction.

## 4. Presentation of the Results

An analysis of the covariance (one-way ANCOVA with Bonferroni correction applied for pairwise comparisons) is carried out to analyze the short-term effects of the wrecked car on the speeds observed. The test contained a total of 43,493 observations. We believe that this incident is relevant since the novelty effect of the speed camera, if there is one, probably faded during this period, and the data analyzed for the pre-test phase are from May 23 to May 30, 2019. The following tables show the number of observations accumulated during the pre-exposure and the exposure of test 1. These observations are normally distributed, and the following Table 1 presents the average velocity of the two phases.

On average, a reduction of 0.17 km/h is noted between the phases of the first test without controlling for confounding variables (i.e., visibility, rain, wind, sunlight, vehicle type, and traffic direction). For the first analysis, we performed a T-test, and it showed a significant reduction between pre-exposure and exposure (*p* < 0.001).

After controlling for the effects of visibility, rain, wind, sunlight, vehicle type, and travel direction, the results still show a significant decrease in speed (*p* < 0.01) from passing vehicles during the wrecked car exposure of 0.637 km/h (see Table 2 below). The significant decrease in speed supports the hypothesis that drivers decrease their driving speed when they are exposed to the wrecked car along the roadside (H1).

From these results, we have retained the observations that exceeded the posted speed limit. During the pre-test phase, we retained 22,029 observations out of the 25,021 accumulated; this number represents 88.04%. With respect to the exposure phase of the damaged car, we find 61,540 observations that exceed the permitted speed limit out of the 71,827 cumulative observations, which represents 85.68%.

The observations that exceed the permitted limit (70 km/h) show an average of 80.91 km/h for the pre-exposure phase and 80.63 km/h for the exposure phase. Without controlling the confounding variables of visibility, rain, wind, sunlight, vehicle type, and traffic direction, a reduction of 0.28 km/h would occur between the two phases of the project. From these results, we continued with an analysis of the covariance (one-way ANCOVA with Bonferroni correction applied for pairwise comparisons), and the following Table 3 shows the results. We can observe that there is a significant difference (*p* < 0.001) between the pre-exposure and exposure phases of 0.277 km/h. This speed reduction is smaller than the one observed for all the observations, which was at 0.637 km/h.

For test 2, we repeated the same experiment but with a before, during, and after design (without group control) consisting of a pre-exposure, an exposure, and a post-exposure in an area where the posted speed limit was 50 km/h. The following Table 4 outlines the sequence of this time series with the total number of observations. In total, 80,225 observations were obtained. These observations are normally distributed, and the following Table 4 presents the average velocity of the three phases.

In general, an average difference of 0.33 km/h is noted between the pre-exposure and the exposure and of 1.24 km/h between the exposure and the post-exposure. A total difference of 1.57 km/h is noted between the pre-exposure and post-exposure phases of the second test. Both the exposure and the post-exposure present significantly slower speeds than the pre-exposure phase (*p* < 0.001).

A one-way ANCOVA with Bonferroni correction applied for pairwise comparisons was performed, and the results show a decrease in vehicle speed of 0.269 km/h between pre-exposure and exposure. The results also show that in the second trial, at the time of post-exposure observation, the vehicles’ speed continued to decrease by 1.255 km/h, which represents a total decrease of 1.524 km/h from the experiment’s start and end phases. The results show that the decrease in speed is significant between these phases (*p* < 0.001) (see Table 5). The pre-exposure, exposure, and post-exposure phases are carried out during the second test. The significant (*p* < 0.001) decrease supports the hypothesis that drivers decrease their speed when they see the wrecked car (H2) and after the car crash exposure (H4).

We performed the same statistical processing during test 1, retaining only the observations that recorded a speed beyond the posted limit (50 km/h) where the test was conducted. The following Table 6 presents the number of observations of the different phases that exceeded 50 km/h during test 2. The number of observations that are above the allowed limit represents 43.25% of the total of the pre-exposure phase, 40.91% of the total of the exposure phase, and 36.14% of the total of the post-exposure phase.

Without the control variables, we can observe a decrease in the average speed during the different phases. Between pre-exposure and exposure, the difference is 0.08 km/h, and from exposure to post-exposure, the decrease is 0.31 km/h. A total decrease of 0.39 km/h is noted for the entirety of this second experiment.

We then performed an analysis of the covariance (one-way ANCOVA with Bonferroni correction applied for pairwise comparisons). The following Table 6 shows the results for observations that recorded a speed above the posted limit. The results of this analysis, which groups the observations recorded at a speed above the permitted limit (>50 km/h), also showed a non-significant decrease between the phases of pre-exposure and exposure. However, the results show that a second significant decrease (*p* < 0.001) of 0.343 km/h is noted between the exposure and post-exposure phases and 0.378 km/h between the pre-exposure and post-exposure phases (H3).

## 5. Discussion of the Results

Considering the high number of speed-related offenses while driving in the City of Saguenay’s territory, the local police have decided to create and deploy an innovative idea of using a damaged car as a means to raise awareness on the consequences of reckless driving, which is the subject of this quasi-experimental before-during-after evaluative study. The hypothesis is that drivers reduce their speed after seeing the wrecked car. To this end, we retained speed as the dependent variable, plus exposure and non-exposure to the activity. The control variables are the direction, type of vehicle (car or truck), visibility, rain, wind, and sunlight. The results show a significant (*p* < 0.001) decrease of 0.637 km/h for the first test and 1.524 km/h for the second test. The results of the second test are encouraging for the police officers of the City of Saguenay since a reduction of 1.6 km/h has been shown to reduce collisions by 5% [5].

A surprising effect is the constant decrease in the speeds recorded during the second test. In fact, during the pre-exposure and exposure phases that lasted 20 days, a significant decrease (*p* < 0.001) was noted of 0.269 km/h, and subsequently, during the following 7 days, a second significant decrease (*p* < 0.001), even more significant than the first, was noted of 1.255 km/h. This assumes that even with 20 days of exposure to the crashed car, the effect persists in drivers. Moreover, this effect persists even after removing it. Thus, there seems to be no saturation effect on driver behaviors related to speed like observed elsewhere [13]. Again, for the second test, after 20 days of exposure to the damaged car and its removal, a significant decrease was analyzed at 0.378 km/h. From this perspective, even for drivers who do not comply with the prescribed limit, the saturation effect is not observed. This advertising campaign succeeds in generating a spontaneous and short-term reaction among exposed drivers, like other campaigns that have been deployed along the roadside [13].

A noticeable effect of accident car exposure on the roadside is related to drivers exceeding speed limits. In fact, when we isolate these observations, during the first test between the pre-exposure and exposure phases, a significant decrease (*p* < 0.001) is revealed at 0.277 km/h in an area where the maximum speed is 70 km/h. During the second test, a significant decrease (*p* < 0.001) was also revealed between the exposure and post-exposure phases at 0.378 km/h, as well as between the pre-exposure and post-exposure phases at 0.343 km/h in an area where the maximum speed is 50 km/h.

Moreover, post-exposure observations are analyzed, and a significant decrease is still observed over time during the second test. Although the results show a low impact, we believe that it is an efficient awareness-raising activity due to the post-exposure effects observed and the low cost associated with such a preventive operation. Conversely, it is not effective given the growing number of speed-related tickets issued each year. However, these awareness campaigns aimed at the general population are necessary to minimize the dual impact on financial and public health that road accidents represent and to modify driving behaviors as much as possible [31,32].

In a similar study, results showed that the use of a realistic-looking police replica placed along the roadside (i.e., Constable Scarecrow) reduced speed along the roadways on which the field study was deployed [13]. As with this project, such interventions represent sustainable avenues in addition to being both inexpensive and easily achievable, plus the police organizations can deploy them regardless of their size and location.

However, it is still impossible to document whether exposing the damaged car creates a source of distraction for drivers when they pass it on the road. Although no collision was documented by the City of Saguenay’s Police Department near the two facilities for the total duration of the data collection carried out, the fact remains that such an event could tarnish the positive effects of the installation observed here. We know that changing messages on the roadside display panels does attract a larger number of observations in addition to a greater percentage of “off-road” observation time when used [33]. It is therefore of the utmost importance that an effort such as the awareness campaign does not become a source of accidents on the road [34]. Moreover, the current study does not allow for differentiation as to which part of the installation—the wreck car, the sign, or both—produced the observed reduction in speed. Future studies should look to differentiate the individual impacts of these parts of the installation.

## 6. Conclusions

To make drivers more aware of the importance of adopting safe driving habits without speeding, the police department of a city in Quebec created and tested a new awareness campaign. The campaign is a wrecked car placed along the roadside. This article supports the hypothesis that motorists reduce their speed when they see a crashed car. Two sets of tests are carried out using statistical radar.

The analysis of the results supports Hypothesis 2 since drivers reduce their speed by 0.637 km/h in the 70 km/h zone and by 0.269 km/h in the 50 km/h zone (significant decrease, *p* < 0.001).

For Hypothesis 3, drivers who exceed the limit in the 70 km/h zone reduce it by 0.277 km/h (a significant decrease, *p* < 0.001). However, for the 50 km/h zone, between the pre-exposure and exposure phases, the speed reduction is not significant, but it is significant between the exposure and post-exposure phases (significant decrease, *p* < 0.001). Thus, Hypothesis 3 is supported.

For Hypotheses 4, The analysis of the results shows a decrease in the speed of all drivers after the exposure of the accident car of 1.255 km/h and of 0.343 km/h for those who do not respect the allowed limit (significant decrease, *p* < 0.001).

The analysis of the results disproves Hypothesis 5 since drivers reduce their speed by 0.269 km/h in the 50 km/h zone and by 0.637 km/h in the 70 km/h zone. However, during exposure and post-exposure, their speed is reduced by 1.524 km/h. Thus, over a longer period, Hypothesis 5 is supported.

These results contribute to the advancement of knowledge concerning a communication strategy relating to the consequences of speeding at the wheel for the general population and for drivers who do not respect the prescribed speed limits. The first theoretical contribution refers to the proposal of Freeman et al. [27] concerning the effectiveness of a realistic awareness campaign, and here, in a real context or on the road

Finally, we can answer our research question about whether the awareness campaign created at low cost by the police of the City of Saguenay has an instantaneous and short-term effect. The answer is affirmative.

The second theoretical contribution refers to driving behaviors that do not respect the prescribed limit and the risk homeostasis described by Wilde [18]. We may be inclined to believe that drivers who tolerate a certain level of risk based on their experience are impervious to a communication strategy aimed at modifying their speeding behaviors at the wheel. Although the results of this research do not specifically track these specific drivers with their license plate number to know if it is still the same person, the results exhibit a significant decrease in the prevalence of higher driving speeds. Furthermore, during the post-exposure phase of the second test, the analysis of the results shows a significant (*p* < 0.001) and constant decrease in speed for all drivers as well as those who do not respect the posted speed limit.

The results of Table 5 present a reflection on the objective targeted by the police officers of the City of Saguenay since they aim to reduce the number of speed-related offenses while driving. Among drivers who exceed the prescribed speed limit, the effects of the campaign are significant but small. However, the results demonstrate the relevance of continuing to reflect on the communication strategy used by local police officers. Moreover, it is relevant to note that there is no saturation effect from this campaign, even if it lasted 20 days, since the analysis of the results shows a noticeable decrease in drivers’ speed, not a stagnation. For future research, it might be relevant to calculate the number of days needed to reach this saturation. As a result, police officers would be able to better plan travel times for this awareness campaign while maximizing each location. Moreover, it would be relevant to achieve a longer post-exposure evaluation period to better know and document the medium-term effects of this campaign. Since other studies of speed-reduction devices lasted for several months and found that the effect may dissipate over time [30,35], such an evaluation period would be beneficial for the current intervention if it is planned to be used for a longer period of time per site.

This research has methodological limitations since there is no control group and the radar can also influence driver behavior, even though it is small and the pre-exposure period is relatively long. In addition, despite the various variables controlled for, other events may have had an effect on driver behavior. In addition, there was no monitoring or tracking of license plates to track the behavior of drivers who violated the speed limit. Finally, the large sample size gathered during the data collection might have contributed to the detection of statistical significance. However, even a small reduction in average driving speed has been shown to be beneficial to road safety [5].

In regard to future research, we suggest conducting interviews with citizens to understand their conscious reactions and the impact of this campaign on their speeding behaviors. This might enlighten the conscious, or unconscious, mechanism underlying the effectiveness of such a campaign.

## Figures and Tables

**Figure 1 ijerph-20-06012-f001:**
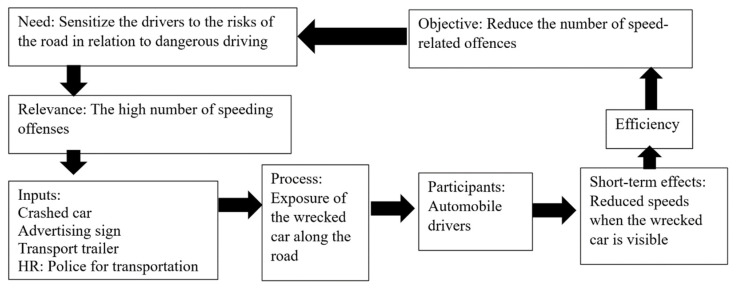
Operational logic of the current awareness campaign.

**Figure 2 ijerph-20-06012-f002:**
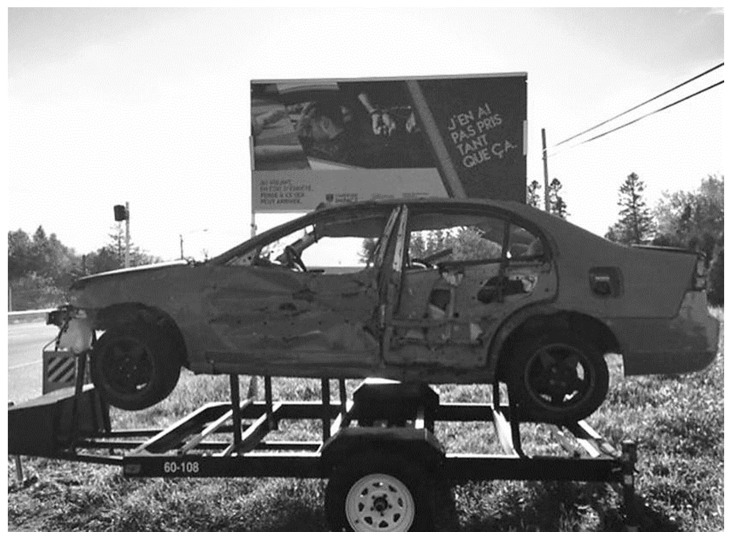
The Saguenay Police Department’s awareness campaign was installed along the region’s roads. The Figure presents the overall installation and the text shown in the background of the wrecked car (“J’en ai pas pris tant que ça”—“I did not take that much”).

**Figure 3 ijerph-20-06012-f003:**
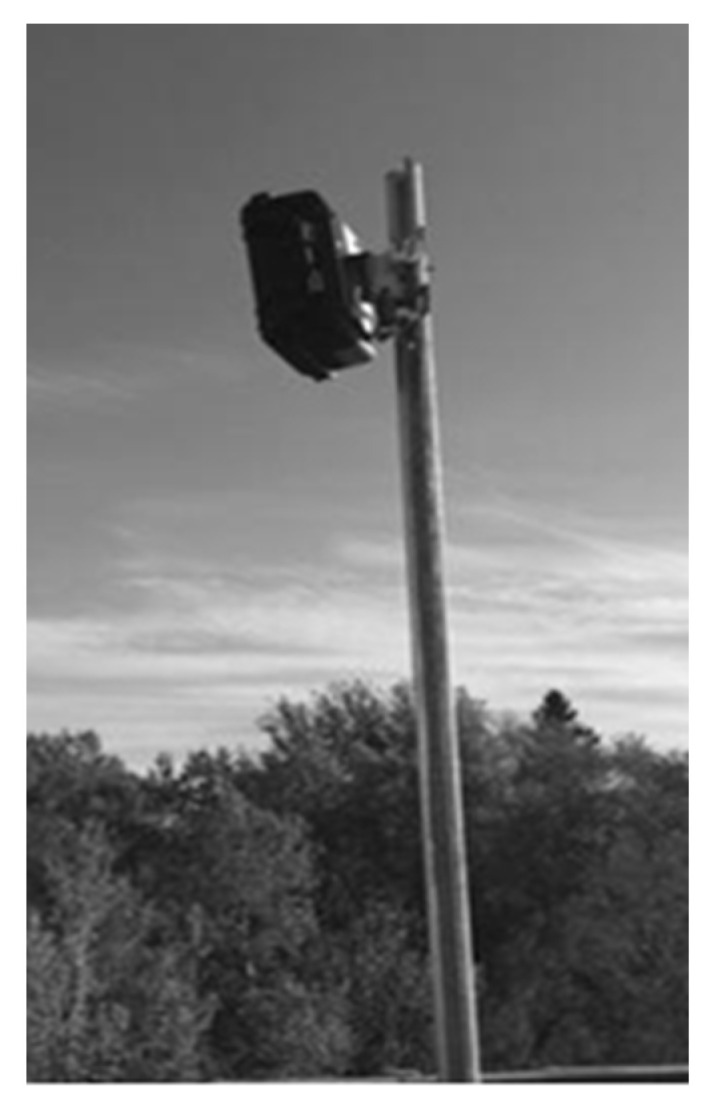
Radar speedometer used in this experiment.

**Figure 4 ijerph-20-06012-f004:**
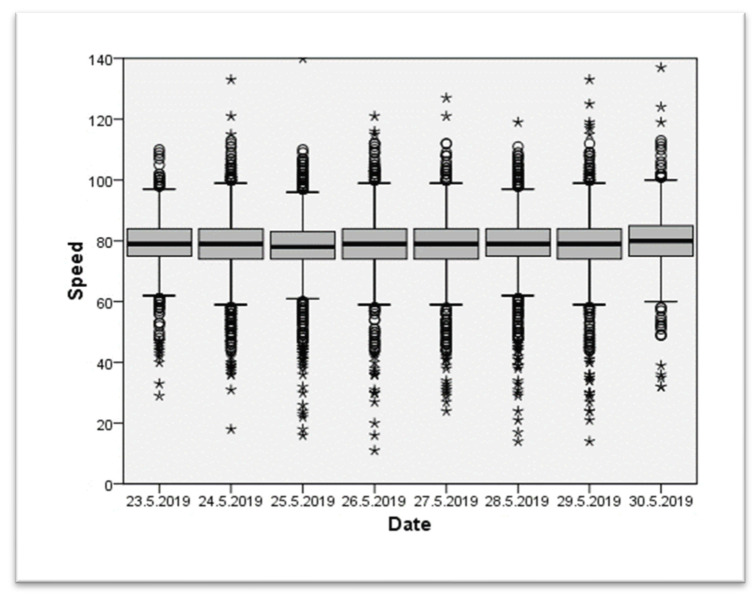
Recorded speed (km/h) during the pre-test of the first data collection in a 70 km/h zone. Outlier data are represented with circle (°) and extreme outlier with *.

**Figure 5 ijerph-20-06012-f005:**
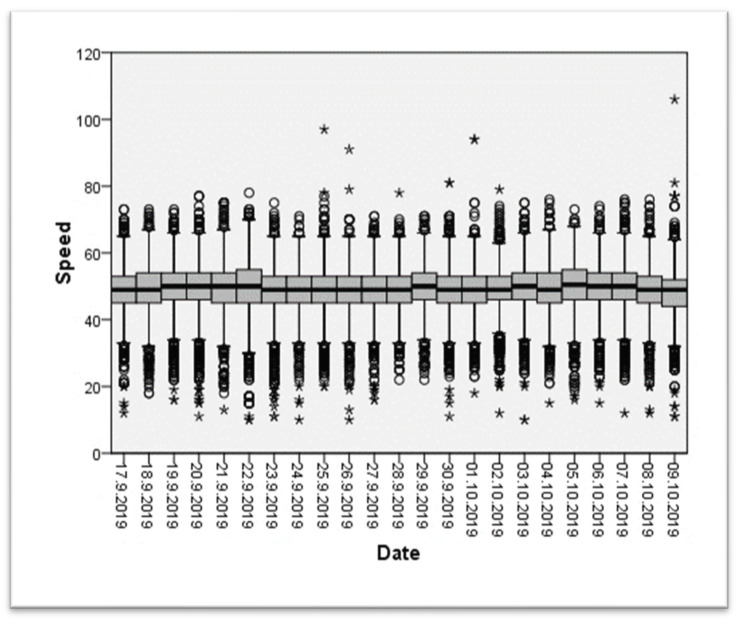
Recorded speed (km/h) during the pre-test of the second data collection in a 50 km/h zone. Outlier data are represented with circle (°) and extreme outlier with *.

**Table 1 ijerph-20-06012-t001:** Average speed (km/h) during the pre-exposure and exposure of test 1.

Phase	Mean	Standard Deviation	N	Days
Pre-exposure May 23 to May 30, 2019	78.90	8.82	25,021	8
Exposure May 31 to June 19, 2019	78.67	8.99	71,827	20
Total	78.73	8.96	96,848	28

**Table 2 ijerph-20-06012-t002:** Results from the first test with pairwise comparisons of speed between the pre-exposure and exposure phases.

(I) Phase	(J) Phase	Mean Difference (I-J)	Std. Error	Sig. ^b^	99.9% Confidence Interval for Difference ^b^	
					Lower Bound	Upper Bound
1 Pre-exposure	2 Exposure	0.637 *	0.066	0.000	0.420	0.853
2 Exposure	1 Pre-exposure	−0.637 *	0.066	0.000	−0.853	−0.420

Based on estimated marginal means. *. The mean difference is significant at the 0.001 level. ^b^. Adjustment for multiple comparisons: Bonferroni.

**Table 3 ijerph-20-06012-t003:** Results from the first test with pairwise comparisons.

(I) Phase	(J) Phase	Mean Difference (I-J)	Std. Error	Sig. ^b^	99.9% Confidence Interval for Difference ^b^	
					Lower Bound	Upper Bound
1 Pre-exposure	2 Exposure	0.277 *	0.052	0.000	0.105	0.450
2 Exposure	1 Pre-exposure	−0.277 *	0.052	0.000	−0.450	−0.105

Based on estimated marginal means. *. The mean difference is significant at the 0.001 level. ^b^. Adjustment for multiple comparisons: Bonferroni.

**Table 4 ijerph-20-06012-t004:** Average speed (km/h) during the pre-exposure, exposure, and post-exposure of test 2.

Phase	Time	Total Number of Days	Mean	Std. Deviation	N
1 Pre-exposure	(September 17 to October 9, 2019)	22	49.14	7.54	48,926
2 Exposure	(October 10 to October 30, 2019)	20	48.81	7.44	22,449
3 Post-exposure	(October 31 to November 6, 2019)	7	47.60	8.00	8850
Total			48.88	7.58	80,225

**Table 5 ijerph-20-06012-t005:** Results from the second test with pairwise comparisons.

(I) Phase	(J) Phase	Mean Difference (I-J)	Std. Error	Sig. ^b^	99.9% Confidence Interval for Difference ^b^	
					Lower Bound	Upper Bound
1 Pre-exposure	2 Exposure	0.269 *	0.063	0.000	0.042	0.497
	3 Post-exposure	1.524 *	0.089	0.000	1.205	1.843
2 Exposure	1 Pre-exposure	−0.269 *	0.063	0.000	−0.497	−0.042
	3 Post-exposure	1.255 *	0.097	0.000	0.905	1.604
3 Post-exposure	1 Pre-exposure	−1.524 *	0.089	0.000	−1.843	−1.205
	2 Exposure	−1.255 *	0.097	0.000	−1.604	−0.905

Based on estimated marginal means. *. The mean difference is significant at the 0.001 level. ^b^. Adjustment for multiple comparisons: Bonferroni.

**Table 6 ijerph-20-06012-t006:** Results from the second test with pairwise comparisons for the number of observations of the different phases that exceeded 50 km/h.

(I) Phase	(J) Phase	Mean Difference (I-J)	Std. Error	Sig. ^b^	99.9% Confidence Interval for Difference ^b^	
					Lower Bound	Upper Bound
1 Pre-exposure	2 Exposure	0.034	0.053	1.00	−0.156	0.225
	3 Post-exposure	0.378 *	0.080	0.000	0.092	0.664
2 Exposure	1 Pre-exposure	−0.034	0.053	1.00	−0.225	0.156
	3 Post-exposure	0.343 *	0.087	0.000	0.033	0.654
3 Post-exposure	1 Pre-exposure	−0.378 *	0.080	0.000	−0.664	−0.092
	2 Exposure	−0.343 *	0.087	0.000	−0.654	−0.033

Based on estimated marginal means. *. The mean difference is significant at the 0.001 level. ^b^. Adjustment for multiple comparisons: Bonferroni.

## Data Availability

The data that supports this study’s findings are available upon request from the corresponding author, M.L.

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
