# Peer review of "Evaluation of a Road Safety Awareness Campaign Deployed along the Roadside in Saguenay (Québec, Canada)"

_ijerph, 2023, doi:10.3390/ijerph20116012_

Round 1

Reviewer 1 Report

1. Some of the descriptions of existing research are not correct and should be fixed before publication. For example, the following statement is not correct: “Therefore, having a collision or getting a ticket is the only outcomes that could alter a driver’s speed.” This sentence should indicate that having a collision or getting a ticket are not necessarily required as research has observed behavioural change in the absence of these consequences.

2. The authors should make it clear that their large sample size likely contributed to the detection of statistical significance in their analyses.

3. The authors should make their limitation of not being able to identify which part of the intervention was most salient for behavioural change more clear in the text.

4. The authors should avoid using causal language like “confirm” and “scientifically proven” given the limitations of their research. Language like “could” and “may” would be more appropriate when describing effects.

5. Some of the references are incorrectly formatted and/or have broken doi hyperlinks.

I believe that this manuscript would benefit from additional proofreading.

Author Response

First and foremost, we would like to thank all three reviewers for their time and suggestions for us the improve the readability and the quality of our manuscript. You will find below all reviewers’ comments, followed by a description of what has been done in the manuscript to address them.

Reviewer 1

  1. Some of the descriptions of existing research are not correct and should be fixed before publication. For example, the following statement is not correct: “Therefore, having a collision or getting a ticket is the only outcomes that could alter a driver’s speed.” This sentence should indicate that having a collision or getting a ticket are not necessarily required as research has observed behavioural change in the absence of these consequences.
    1. The sentence was changed as per the reviewer’s suggestion. Thanks for highlighting this incorrectness.

  1. The authors should make it clear that their large sample size likely contributed to the detection of statistical significance in their analyses.
    1. A note was added in the limitation section of the paper in this regard. It has to be noted that other studies have shown that even such a slight reduction in speed has been shown beneficial for road safety (CCMTA, 2013). Therefore, such results are interesting in terms of preventing collisions.

  1. The authors should make their limitation of not being able to identify which part of the intervention was most salient for behavioural change more clear in the text.
    1. A note was added accordingly at the end of the paragraph after line 340.

  1. The authors should avoid using causal language like “confirm” and “scientifically proven” given the limitations of their research. Language like “could” and “may” would be more appropriate when describing effects.
    1. Thanks for the suggestion. We have modified the manuscript accordingly.

  1. Some of the references are incorrectly formatted and/or have broken doi hyperlinks.
    1. There seems to have been an issue when the original Word manuscript was transformed to the PDF proof since all DOI have been manually verified before submission, and they all worked correctly. We took the time to verify them once again one by one, and they all work in the Word document that serves for the submission. We are sorry to hear that they were not working correctly in the pdf version.

  1. I believe that this manuscript would benefit from additional proofreading.

We had the text proofread by a native English speaker. If ever it is possible, we would be happy to have the text revised once again by a third party.

Reviewer 2 Report

The authors of the article present research focused on the safety of vehicle operation. Based on the installation of a crashed car by the road, they assess whether it will have an impact on reducing the speed on selected sections of the road. For the purposes of the experiment, road sections with a maximum permitted speed of 70 km/h and 50 km/h were selected. As part of the research, the authors correctly took into account other influences that could affect the speed of vehicles (rain, wind, etc.). The experimental data are statistically evaluated and show the positive impact of installing a crashed car on reducing the speed of cars. I believe that the research is presented in a comprehensible manner and the results are beneficial for the field of traffic safety and thus the negative impact of traffic accidents on the environment is also eliminated.

I have only the following minor comments on the article:

- please better formulate the abstract "The results show a significant decrease (99.9%) of 0.637 km/h for the first test and 0.269 km/h for the second test when the device is exposed." The reader will not know what is meant by the first test and what is meant by the second test. I don't understand the value of 99.9%.

- it is necessary to enlarge image 2, furthermore, I did not understand from the image and the text whether the billboard behind the car is related to the experiment

- the literature does not comply with the autoty guidelines (the journal requires numerical citations).

In addition to the scientific contribution, the article also has a practical contribution.

Author Response

First and foremost, we would like to thank all three reviewers for their time and suggestions for us the improve the readability and the quality of our manuscript. You will find below all reviewers’ comments, followed by a description of what has been done in the manuscript to address them.

Reviewer 2

The authors of the article present research focused on the safety of vehicle operation. Based on the installation of a crashed car by the road, they assess whether it will have an impact on reducing the speed on selected sections of the road. For the purposes of the experiment, road sections with a maximum permitted speed of 70km/h and 50 km/h were selected. As part of the research, the authors correctly took into account other influences that could affect the speed of vehicles (rain, wind, etc.). The experimental data are statistically evaluated and show the positive impact of installing a crashed car on reducing the speed of cars. I believe that the research is presented in a comprehensible manner and the results are beneficial for the field of traffic safety and thus the negative impact of traffic accidents on the environment is also eliminated.

                We thank the reviewer for this comment.

I have only the following minor comments on the article:

- please better formulate the abstract "The results show a significant decrease (99.9%) of 0.637 km/h for the first test and 0.269 km/h for the second test when the device is exposed." The reader will not know what is meant by the first test and what is meant by the second test. I don't understand the value of 99.9%.

The word test has been changed to “site” in order to differentiate both for the reader better. We also added the speed limit for each site.

The value of 99.9% was for the p-value. The abstract has been changed to p<0,001.

- it is necessary to enlarge image 2, furthermore, I did not understand from the image and the text whether the billboard behind the car is related to the experiment

Figure 2 has been modified to highlight the sign behind the wrecked car better. Figure 2 now presents a left panel with the entire set-up and a right panel with a zoom on the message in the background that is part of the campaign.

- the literature does not comply with the authors guidelines (the journal requires numerical citations).

In the original submission, references were left in the APA format for readability, but all we’ll be changed to numerical citations once the paper is accepted.

In addition to the scientific contribution, the article also has a practical contribution.

Thanks. This is the reason why we see the importance behind this work from both a scientific standpoint but also on its practicability for police organizations.

Reviewer 3 Report

The paper is interesting and topical - speed and speeding is clearly a number one risk factor in traffic safety, which calls for practical studies and effective interventions.

However, I have some comments on the paper:

Firstly, it is stated that the campaign focused on "those who do not respect the speed limits", specifically on experienced male drivers. But the applied design did not allow targetting specific drivers - instead the speed of all drivers is measured and analyzed. Thus, the link between the objective and the results should be made more clear.

Secondly, the identified differences, although statistically significant, are in reality very small - below 1 km/h. I even wonder if they are not caused by some technical inaccuracy or a random variation from uncontrolled variables. What is the real-life value of such small speed reduction? This should be discussed and explained, including comparison with speed reduction from other similar studies.

Thirdly, the 20-day post-exposure period is quite short. Other studies of speed-reduction devices lasted for several months and found that the effect may deteriorate in time (e.g., Lee et al., 2006; Gates et al., 2008; Gehlert et al., 2012). Again, this should be explained and discussed.

References:

Lee et al. (2006). Effectiveness of Speed-Monitoring Displays in Speed Reduction in School Zones. TRR.

Gates et al. (2008). Effectiveness of Experimental Transverse-Bar Pavement Marking as Speed-Reduction Treatment on Freeway Curves. TRR.

Gehlert et al. (2012). Evaluation of different types of dynamic speed display signs. TR-F.

Abbreviations (such as "it's") should not be used in an academic text.

Author Response

First and foremost, we would like to thank all three reviewers for their time and suggestions for us the improve the readability and the quality of our manuscript. You will find below all reviewers’ comments, followed by a description of what has been done in the manuscript to address them.

Reviewer 3

The paper is interesting and topical - speed and speeding is clearly a number one risk factor in traffic safety, which calls for practical studies and effective interventions.

                Thanks.

However, I have some comments on the paper: Firstly, it is stated that the campaign focused on "those who do not respect the speed limits", specifically on experienced male drivers. But the applied design did not allow targeting specific drivers - instead the speed of all drivers is measured and analyzed. Thus, the link between the objective and the results should be made more clear.

A note is made in this regard in the paragraph at line 370. Although we agree with the reviewer’s comment that the intervention can not clearly differentiate on whom it was successful in terms of reducing driving speed, the results show that the higher speed above the posted speed limit was reduced with the intervention. Thus, we have changed “drivers” for driving behaviours in the initial objective.

Secondly, the identified differences, although statistically significant, are in reality very small - below 1 km/h. I even wonder if they are not caused by some technical inaccuracy or a random variation from uncontrolled variables. What is the real-life value of such small speed reduction? This should be discussed and explained, including comparison with speed reduction from other similar studies.

As suggested by Reviewer 1, a note has been made in the manuscript to highlight this potential bias since the sample size is large. However and as previously mentioned, even slight reductions in average driving speed have been shown to be beneficial for road safety (CCMTA, 2013).

Thirdly, the 20-day post-exposure period is quite short. Other studies of speed-reduction devices lasted for several months and found that the effect may deteriorate in time (e.g., Lee et al., 2006; Gates et al., 2008; Gehlert et al., 2012). Again, this should be explained and discussed. References: Lee et al. (2006). Effectiveness of Speed-Monitoring Displays in Speed Reduction in School Zones. TRR., Gates et al. (2008). Effectiveness of Experimental Transverse-Bar Pavement Marking as Speed-Reduction Treatment on Freeway Curves. TRR., Gehlert et al. (2012). Evaluation of different types of dynamic speed display signs. TR-F.

As per the reviewer’s suggestion, we added a section around line 394 in this regard. Thanks for the references suggested here. They were helpful in supporting this point in the manuscript.

Abbreviations (such as "it's") should not be used in an academic text.

                The text has been modified to prevent the use of this abbreviation.

Round 2

Reviewer 3 Report

Thank you for the revision. I can see that the review comments were addressed and the paper quality has improved. I recommend the paper to be accepted for publication in the journal.